# Diagnostic Value of Inflammatory Markers in Elderly Arab Women with Sarcopenia

**DOI:** 10.3390/diagnostics12102293

**Published:** 2022-09-23

**Authors:** Dara Aldisi, Mahmoud M. A. Abulmeaty, Tafany A. Alsaawi, Abeer S. Alorf, Gadah Mujlli, Atheer M. Alshahrani, Rajwa M. Alahmari, Mohammed Alquraishi, Nasser M. Al-Daghri, Nawaf W. Alruwaili, Shaun Sabico

**Affiliations:** 1Department of Community Health Sciences, College of Applied Medical Sciences, King Saud University, Riyadh P.O. Box 11362, Saudi Arabia; 2Biochemistry Department, College of Science, King Saud University, Riyadh P.O. Box 11451, Saudi Arabia

**Keywords:** sarcopenia, TNF-α, IL-6, Arab elderly women, inflammation

## Abstract

Several studies have found a correlation between inflammatory markers and sarcopenia; however, limited research has been conducted on the Arabic population. Therefore, this study aimed to investigate the value of inflammatory parameters in Saudi elderly women with sarcopenia. In this cross-sectional study, 76 elderly Saudi women (>65 years) were stratified according to the presence (*n* = 26) or absence (*n* = 50) of sarcopenia, using the operational definition of the Asian Working Group for Sarcopenia (AWGS). Demographics and clinical data were collected. Muscle mass, muscle strength, and physical performance were assessed using bioelectrical impedance, hand grip and timed-up-and-go (TUG) tests, respectively. Inflammatory markers such as interleukin-6 (IL-6), tumor necrosis factor alpha (TNF-α) and C-reactive protein (CRP) were assessed using commercially available assays. Muscle mass and strength indicators were lower in the sarcopenia group (*p*-value < 0.05). Moreover, interleukin 6 (IL-6) was positively correlated with TUG (r = 0.48, *p*-value < 0.05), while CRP showed an inverse correlation with the right leg muscle (R-Leg-M) and a positive correlation with triceps skinfold (TSF) (r = −0.41, r = 0.42, respectively, *p*-values < 0.05). Additionally, TSF and R-Leg-M were independent predictors of CRP variation (R^2^ = 0.35; *p* < 0.01). Lastly, participants with a TNF-α > 71.2 were five times more likely to have sarcopenia [(OR = 5.85), 95% CI: 1.07–32.08; *p* = 0.04]. In conclusion, elevated levels of TNF-α are significantly associated with the risk of sarcopenia, while variations perceived in circulating CRP can be explained by changes in the muscle mass indices only among individuals with sarcopenia. The present findings, while promising, need further investigations on a larger scale to determine whether inflammatory markers hold any diagnostic value in assessing sarcopenia among elderly Arab women.

## 1. Introduction

Sarcopenia is prevalent in most settings of the elderly population. It is estimated that the prevalence of sarcopenia is as high as 29% among community-living, elderly women, 14–33% among institutionalized men, and 10% among hospitalized individuals [1]. Sarcopenia is currently defined as an age-associated loss of skeletal muscle mass, reduced muscle strength, and/or physical performance associated with diminished physical capability, disability, impaired cardiopulmonary performance, and mortality [2]. Sarcopenia has gathered increasing interest among clinicians over recent decades and is now categorized as a geriatric syndrome together with falls and dementia, to name a few [3]. In 2010, the European Working Group on Sarcopenia in Older People (EWGSOP) proposed a widely used diagnostic strategy for sarcopenia [2]. This strategy reported low muscle mass and low muscle function, strength, or performance as a diagnostic measure for sarcopenia [2]. The same strategy, with different cut-off points, was developed in 2014 by the Asian Working Group for Sarcopenia (AWGS) [4]. The AWGS established a consensus on sarcopenia diagnosis, which defined a set of approaches for measuring muscle mass, muscle strength, and physical performance, plotted on different cut-off values based on Asian populations [4].

A stage of severe sarcopenia is recognized by all three criteria of the definition. Therefore, identifying the stages of sarcopenia might determine management modalities and goals [2]. It has been suggested that systemic inflammation associated with aging may contribute to the progression of sarcopenia [5]. However, the suggested mechanisms are complex, and it is still difficult to establish a full biochemical explanation. Recent studies have shown that inflammatory cytokines could promote muscle-wasting by enhancing protein catabolism and inhibiting muscle synthesis [5,6,7]. In fact, a recent meta-analysis of 17 cross-sectional studies involving a total of 11,249 participants confirmed that those with sarcopenia had significantly higher CRP levels than those without, while serum IL-6 levels and TNF-α were comparable in both groups [8].

In an Italian cross-sectional study, the serum level of reduced glutathione (GSH) was significantly lower in patients with sarcopenia compared to the controls. At the same time, no difference was detected between the two groups regarding oxidized glutathione GSSG levels. Nevertheless, GSSG/GSH in patients with sarcopenia was increased, and malondialdehyde (MDA) and 4-hydroxy-2,3-nominal (HNE) protein adducts could predict the presence of sarcopenia [9]. Another study that included 107 pre-and postmenopausal women found an inverse correlation between the lipid peroxide level and skeletal muscle index, suggesting that levels of lipid peroxide may partially explain the decrease in skeletal muscle mass among postmenopausal women [10].

Given the limited research conducted among the ethnic Arabian population, and the clinical implications of sarcopenia as the community ages, this study aimed to investigate the diagnostic value of inflammatory parameters in elderly Arabic women with sarcopenia living in Riyadh, Saudi Arabia.

## 2. Materials and Methods

### 2.1. Study Design and Setting

This present multi-center, cross-sectional study was conducted in two primary health care centers, Aldiriyah and Alsalam centers, and two community centers (King Salman social center and Quran memorizing centers) in Riyadh, Saudi Arabia. Participants visiting these study locations were recruited and invited via text messages or face-to-face communication. Each consenting participant was required to visit once for data collection.

### 2.2. Study Participants

The study’s inclusion criteria were elderly Saudi women aged between 60 and 85 years old. Figure 1 shows the flowchart of participants. For the purpose of this study, sarcopenia cases were based on the operational definition provided by the AWGS, defined as low muscle mass (<5.7 kg/m^2^) with low muscle strength (handgrip < 18 kg) or low physical performance (TUG < 20 s), as described previously [3,11]. The exclusion criteria included those who could not move without assistive devices such as canes or walkers; those who had a history of chronic diseases including chronic obstructive pulmonary disorder (COPD), congestive heart failure (CHF), chronic renal failure (CRF), cirrhosis or liver failure, and active cancer; those who had artificial limbs or limb prostheses, and those with poorly controlled medical problems. The study was explained to all participants, and consent was collected prior to their inclusion. The study was approved by the Institutional Review Board (IRB) at the College of Medicine, King Saud University (Log No. E-19-368).

### 2.3. Demographic Data and Medical Information 

A questionnaire was used to collect the participant’s demographic and medical information. The demographic data included: age, socioeconomic status, educational level, income rate, and marital status. The medical information included: medical history, smoking, and current medications.

### 2.4. Anthropometric Measurements

Trained nurses assessed the anthropometric measurements for each participant, including waist circumference (WC), hip circumference (HC), weight, height, and midarm muscle area (MAMA). The WC (cm) was measured twice at the midpoint between the tenth rib’s lower border and the iliac crest, and the average was noted. The HC (cm) was measured twice around the widest portion of the buttocks, and the average was noted [12]. BMI was calculated as kg/m^2^. MAMA was calculated according to this equation: “{MAMA = (MAC − π × TSF)2/4π}”; MAC is midarm circumference, and TSF is triceps skinfold thickness [13]. Conicity index was determined by the following formula: “[(CI = WC (m)/[0.109 × √{weight (kg)/Height (m)}]” [12]. Abdominal volume index (AVI) was calculated according to the following equation: “AVI = [2 × (WC)2 + 0.7 × (waist − hip)2]/1000” [11,14].

### 2.5. Muscle Mass Quantification

Bioelectrical impedance (BIA, Tanita BC-418, Tanita Co, Tokyo, Japan) was used to assess body composition, especially total and segmental muscle masses, which were used for analysis. The values of the skeletal muscle mass were derived from the predicted muscle mass equation = “[(Ht2/R × 0.401)1 + (gender × 3.825) + (age × −0.071)] + 5.102”. “R” represents the BIA resistance (ohms), while sex has a score of zero for females and one for males [15]. According to the AWGS, females with a muscle mass < 6.4 kg/m^2^ were considered to have low muscle mass [4].

### 2.6. Muscle Strength

The handgrip strength (HGS) by using a hydraulic hand dynamometer (Lafayette Instruments Co., Lafayette, IN, USA) was used to measure muscle strength. Participants were asked to squeeze the hydraulic dynamometer with their right and left hands, noting the average measurement. Low handgrip strength is defined as <18 kg for women by the AWGS [4].

### 2.7. Physical Performance

The participant’s muscle performance was evaluated using the three meters timed-up-and-go test (TUG). TUG was conducted by asking the participant to sit in a chair, rise and walk at average speed for 3 m, and then go back to the same chair while measuring the time required to complete the task using a stopwatch. A speed of ≤20 s was considered an indicator of low muscle performance, based on EWGSOP recommendations [16].

### 2.8. Biochemical Analyses

A certified phlebotomist withdrew about five ml of fasting blood samples from each participant for biochemical analysis. All blood samples were centrifuged (3000 RPM for 10 min) and stored in a −80 °C freezer before the analysis. Serum fasting glucose and lipids were analyzed routinely using a chemical analyzer (Konelab, Vintaa, Finland). Circulating levels of CRP, IL-6 levels, and TNF-α were measured using commercially available assay kits following the manufacturer’s protocol (MyBioSource, San Diego, CA, USA; catalog numbers: MBS2505217, MBS8123859, and MBS508481, respectively) from previous observations [17]. Intra- and inter-assay CVs were as follows: CRP (3.95%, 6.07%), IL-6 (2.73%, 2.6%) and TNF-α (6.3%, 6.4%). All biochemical analyses were performed at the Chair for Biomarkers of Chronic Diseases (CBCD), King Saud University, Riyadh, Saudi Arabia.

### 2.9. Sample Size Calculation and Data Analyses

The sample size was obtained from previous examples in the literature, comparing inflammatory cytokines among Iranian adults with and without sarcopenia and noting increased CRP levels among cases with an effect size of 0.92 [18]. The calculation was performed using G*Power ((G*Power 3.1.9.4, Heinrich Heine University, Dusseldorf, Germany) where α = 0.05 and power = 0.9; the estimated sample size was N = 22 per group.

Statistical analysis was performed using SPSS (version 25, Chicago, IL, USA). Categorical characteristics were shown as frequency and percentages, while continuous data were shown as mean and ± standard deviation (SD). Crosstabs with a chi-square test compared the categorical variables (demographic parameters). An independent Student T-test or Mann–Whitney U test was used to compare differences between participants with and without sarcopenia. Stepwise linear regression analysis was performed to determine significant predictors for TNF-α, IL-6, and CRP. Binary logistic regression analysis was used to assess the tertiles of inflammatory markers with a risk of sarcopenia. A *p*-value < 0.05 was considered statistically significant.

## 3. Results

### 3.1. Clinical Characteristics of Participants

The study included 76 participants (*n* = 50 women without sarcopenia and *n* = 26 with sarcopenia). Table 1 shows the demographic data of the study groups. Weight, BMI, WC, and HC were significantly lower in the sarcopenia group (*p*-values < 0.05) (Table 2).

Additionally, muscle strength and function indicators, including MAC, MAMA, AVI, HGS, muscle mass, R-leg-M, R-arm-M, L-arm-M, trunk, and muscle mass, by the predicted muscle mass equation, were lower in the sarcopenia group (*p*-value < 0.05) (Table 3). 

The biochemical analysis of the study participants showed no significant differences in glucose, lipid profile, and cytokine concentrations between the groups (Table 4). 

### 3.2. Correlation between Inflammatory Markers and Study Parameters

The Spearman correlation analysis between the inflammatory markers (IL-6, TNF-α, CRP) and other study parameters are presented in Table 5. No significant associations were observed between the inflammatory markers and studied parameters when all participants were used (not shown in the table). However, when stratified according to sarcopenia status, IL-6 was positively correlated with UGT among the sarcopenia group (r = 0.48, *p*-value < 0.05), and negatively correlated with TNF-α among the control group, respectively (r = −0.40, r = −0.45, *p*-value < 0.01) (Table 5). In addition, CRP showed an inverse correlation with the R-leg-M (r = −0.41, *p*-value < 0.05) and a positive correlation with triceps skinfold thickness (r = 0.42, *p*-value < 0.05) in the sarcopenia group.

Logistic regression analysis was carried out to associate the inflammatory markers IL-6, TNF-α, and CRP with sarcopenia (Table 6), and it was found that TNF-α was the only significant predictor of sarcopenia in tertile 3 or at levels > 71.2 [(OR = 5.85), 95% (1.07–32.08)].

Stepwise regression analysis was conducted to determine which of the study parameters would be a predictor of variation in the inflammatory markers among the study groups. Among the sarcopenia group, TSF and R-Leg-M were independent predictors of CRP, explaining 35% of the variances perceived (R^2^ = 0.35) in the sarcopenia group [(0.04(0.01), *p* = 0.005), (−0.15 (0.06), *p* = 0.025; respectively]. No independent predictors for IL = 6 were observed. Lastly, MAMA was an independent predictor of TNF-α, explaining 9% of the variances perceived among those without sarcopenia (R^2^ = 0.09; *p* < 0.05). 

## 4. Discussion

Sarcopenia is the age-associated loss of skeletal muscle mass that affects the quality of life of the elderly population [1,2]. The present cross-sectional study aimed to identify the influence of inflammatory markers as potential diagnostic markers for sarcopenia in a group of elderly Arab women. The study found no differences in the serum levels of CRP, TNF-α, and IL-6 between the study groups. Bano et al. [8] reported that participants with sarcopenia had significantly higher levels of CRP than the controls, while the levels of TNF-α and IL-6 were insignificantly different. Moreover, Asoudeh et al. [18] found that the levels of CRP, IL-6, and TNF-α were not significantly different between participants with and without low muscle masses as well as those with and without abnormal gait speed tests. However, circulating CRP was elevated among those with a low HGS. In the Copenhagen sarcopenia study, elderly women also had significantly higher levels of CRP, TNF-α, and IL-6, compared with middle-aged women and significantly higher levels of CRP, TNF-α, IL-1β, and IL-4 compared with young girls [19]. On the other hand, Rong et al. [20] reported that IL-6 and IL-10 concentrations, as well as the IL-6/IL-10 ratio, were significantly higher in the elderly sarcopenia group than in those without sarcopenia.

In the present study, IL-6 was positively correlated with TUG, while CRP was positively correlated with TSF and inversely correlated with R-leg-M in the sarcopenia group only. This observation again supplements the findings from the Copenhagen sarcopenia study, where the association of CRP with a lower appendicular lean mass as well as with lower handgrip strength was evident in elderly women in particular [19]. A similar study observed an association between sarcopenia with CRP, ESR, and adiponectin [21]. By contrast, Asoudeh and her colleagues reported no correlations between IL-6, TNF-α, and CRP with sarcopenia [18]. Furthermore, Dupont et al. [22] reported that baseline levels of inflammatory markers such as CRP, white blood cell count, and albumin did not predict sarcopenia in elderly men. This lack of consistency in the associations between inflammatory cytokines and sarcopenia can be explained by the differences in sample size, age, sex, and ethnic background of participants, not to mention the different assays used.

Another highlight in the present study was that in the sarcopenia group, both TSF and R-Leg-M predicted significant variations in CRP levels, while MAMA explained significant variations in TNF-α levels. The elevated levels of TNF-α at levels higher than 70 ng/mL also translated to a higher risk of sarcopenia. These findings support the cumulative effects of inflammation in the progression of sarcopenia, as inflammation promotes a loss of muscle mass, strength, and function via the modulation of both muscle protein breakdown and synthesis through several signaling pathways [23]. In a recent 3-year longitudinal study investigating the changes in inflammatory markers among the elderly population, higher levels of IL-6 were associated with an increased decline in muscle strength by −3.21 ± 0.81 kg. It was also found that elevated IL-6 and CRP levels were associated with a 2–3-fold risk of losing more than 40% muscle strength [24]. Altered inflammatory markers in the elderly have also been investigated through multivariate modeling or a multi-marker approach and have shown that a combination of signature biomarkers, which include inflammation and muscle modeling, may partially explain the complex pathophysiology of sarcopenia as well as frailty in the elderly [25]. This clustering or ‘omics’ approach may shed more light on the current understanding of sarcopenia as it can cover wider systems not limited to inflammation, such as genetics and nutrition, all of which are still relatively understudied in this field [26]. 

The authors can acknowledge several limitations. The causal relationship between inflammatory markers and sarcopenia cannot be determined, given the study design. Furthermore, the medical history provided by participants was not verified in medical records. Hence, the accuracy of information may be subject to recall bias. Participants were also limited to women, and, as such, these findings could not be generalized. Lastly, although the sample size was adequate, adjustments for potential confounders not considered in the present study, such as physical activity and medications, were not conducted as it may have increased the likelihood of type 2 errors. Despite these limitations, this study is the first to associate inflammatory markers with sarcopenia among elderly Arab women. Interventional studies that can characterize the behavior of inflammatory markers among women with sarcopenia undergoing treatment can verify the present findings. 

## 5. Conclusions

An elevated TNF-α is associated with a higher risk of sarcopenia, while variances perceived in other inflammatory markers, such as CRP, are associated with select muscle indices found only among elderly women with sarcopenia. These findings, while preliminary, open up the possibility for further investigations on the role of inflammatory markers and body composition among elderly Arab women. Further clinical research should be conducted on the validity and diagnostic value of this association.

## Figures and Tables

**Figure 1 diagnostics-12-02293-f001:**
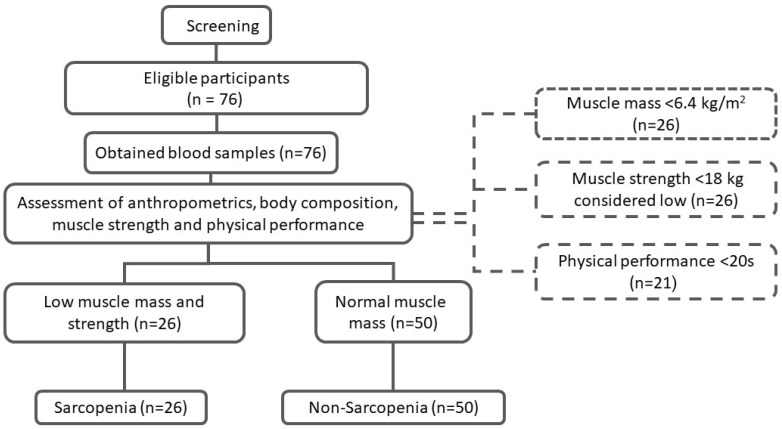
Screening process and enrolment of study participants.

**Table 1 diagnostics-12-02293-t001:** Demographic characteristics of participants according to sarcopenia status.

	All (*n* = 76)	Non-Sarcopenia (*n* = 50)	Sarcopenia (*n* = 26)	*p*-Value
Education	39 (51.3)	25 (50.0)	14 (53.8)	0.93
Illiterate				
Elementary	20 (26.3)	13 (26.0)	7 (26.9)	
Middle school	7 (9.2)	5 (10.0)	2 (7.7)	
High School	3 (3.9)	2 (4.0)	1 (3.8)	
College degree	7 (9.2)	5 (10.0)	2 (7.7)	
Marital Status Married	50 (65.8)	34 (68.0)	16 (61.5)	0.42
Widowed	24 (31.6)	14 (28.0)	10 (38.5)	
Divorced	2 (2.6)	2 (4.0)	0	
Employment Unemployed	66 (86.8)	43 (86.0)	23 (88.5)	0.58
Retired	8 (31.6)	5 (10.0)	3 (11.5)	
Home Business	2 (2.6)	2 (4.0)	0	
Medical history DM	46 (60.5)	29 (58.0)	17 (65.4)	0.36
Hypertension	54 (71.1)	36 (72.0)	18 (69.2)	0.50
High Cholesterol	35 (46.1)	25 (50.0)	10 (38.5)	0.24
Osteoporosis	6 (7.9)	4 (8.0)	2 (7.7)	0.67
Rheumatoid	2 (2.6)	1 (2.0)	1 (3.8)	0.57
Asthma	6 (7.9)	5 (10.0)	1 (3.8)	0.32
Hypothyroidism	6 (7.9)	6 (12.0)	0	0.07
Comorbidity	54 (71.1)	37 (74.0)	17 (65.4)	0.30
One health condition	69 (90.8)	47 (94.0)	22 (84.6)	0.18

Note: Data presented as *n* (%). *p*-value significant at <0.05.

**Table 2 diagnostics-12-02293-t002:** Differences in age and anthropometrics according to sarcopenia status.

Parameters	All	Non-Sarcopenia	Sarcopenia	*p*-Value
*n*	76	50	26	
Age (years)	66.8 ± 5.7	66.5 ± 5.7	67.5 ± 5.7	0.11
BMI (kg/m^2^)	30.7 ± 5.6	32.2 ± 5.8	27.8 ± 2.7	0.001
Waist (cm)	92.8 ± 11.3	95.8 ± 11.2	87.5 ± 9.7	0.003
Hips (cm)	106.8 ± 10.4	109.0 ± 11.4	102.7 ± 6.6	0.02
WHR (cm)	0.87 ± 0.07	0.87 ± 0.08	0.86 ± 0.07	0.54

Note: Data presented as mean ± SD. Significant at *p* < 0.05.

**Table 3 diagnostics-12-02293-t003:** Muscle mass, strength, and performance according to sarcopenia status.

Parameters	All	Non-Sarcopenia	Sarcopenia	*p*-Value
*n*	76	50	26	
Midarm Circumference	28.0 ± 3.7	29.0 ± 3.6	26.2 ± 3.1	0.002
Triceps Skinfold Thickness	17.3 ± 3.4	17.5 ± 3.2	16.8 ± 3.2	0.43
Conicity Index	1.24± 0.11	1.24 ± 0.11	1.23 ± 0.12	0.77
Midarm Muscle Area	41.4 ± 11.5	44.5 ± 11.7	35.3 ± 8.5	0.001
Abdominal Volume Index	17.3 ± 4.4	18.3 ± 4.5	15.5 ± 3.4	0.009
Handgrip Strength	15.7 ± 4.3	16.9 ± 4.3	13.4 ± 3.4	0.001
TUG	16.2 ± 3.9	16.3 ± 4.1	16.0 ± 3.4	0.75
Muscle Mass	39.1 ± 4.7	40.8 ± 4.7	35.9 ± 2.8	<0.001
Right Leg Muscle	6.6 ± 1.0	6.8 ± 1.0	6.1 ± 0.7	0.001
Left Leg Muscle	6.7 ± 1.2	6.9 ± 1.0	6.4 ± 1.4	0.11
Right Arm Muscle	1.9 ± 0.3	2.0 ± 0.3	1.7 ± 0.2	<0.001
Left Arm Muscle	2.0 ± 0.3	2.1 ± 0.3	1.8 ± 0.2	<0.001
Trunk	21.9 ± 2.7	23.0 ± 2.5	19.9 ± 1.9	<0.001
Predictive equation (kg/m^2^)	6.5 ± 0.9	6.9 ± 0.9	5.9 ± 0.3	<0.001

Note: Data presented as mean ± SD. *p*-value significant at <0.05.

**Table 4 diagnostics-12-02293-t004:** Biochemical analysis according to sarcopenia status.

Parameters	All	Non-Sarcopenia	Sarcopenia	*p*-Value
*n*	76	50	26	
Glucose (mmol/L)	10.8 ± 4.0	10.6 ± 3.8	11.0 ± 4.6	0.75
Total Cholesterol (mmol/L)	5.2 ± 1.0	5.1 ± 1.1	5.3 ± 1.1	0.70
HDL-Cholesterol (mmol/L)	1.5 ± 0.4	1.5 ± 0.4	1.4 ± 0.4	0.70
CRP	3.0 (1.5–5.2)	2.7 (1.4–5.0)	3.6 (1.8–5.6)	0.19
IL-6	9.9 ± 2.5	10.1 ± 2.6	9.72 ± 2.1	0.59
TNF-α	30.3 (5.6–75.5)	27.5 (3.4–73.4)	50.8 (17.4–78)	0.13

Note: Data presented as mean ± SD, median (25th–75th percentile).

**Table 5 diagnostics-12-02293-t005:** Correlation analysis for IL-6, TNF-α, and CRP with other parameters.

Parameters	IL-6				Log-TNF-α				Log-CRP			
	Non-Sarc		Sarcopenia		Non-Sarc		Sarcopenia		Non-Sarc		Sarcopenia	
	r	*p*	r	*p*	r	*p*	r	*p*	r	*p*	r	*p*
Age (years)	0.17	0.26	0.19	0.36	−0.20	0.18	−0.40	0.06	0.03	0.86	0.27	0.18
Body Composition												
BMI (kg/m^2^)	−0.12	0.42	−0.15	0.47	0.19	0.21	0.17	0.44	0.15	0.28	0.13	0.53
Waist (cm)	−0.17	0.28	−0.01	0.98	0.29	0.06	−0.12	0.61	0.04	0.79	0.26	0.23
WHR	−0.20	0.20	0.31	0.15	−0.05	0.74	−0.29	0.20	−0.02	0.91	0.28	0.19
Glucose and Lipids												
Glucose (mmol/L)	−0.06	0.80	0.18	0.44	0.06	0.60	0.12	0.44	0.05	0.93	0.03	0.96
Total Chol (mmol/L)	0.05	0.76	0.08	0.73	−0.18	0.28	0.02	0.92	0.07	0.66	0.10	0.67
HDL-Chol (mmol/L)	−0.10	0.55	−0.05	0.81	0.00	0.98	−0.16	0.51	−0.07	0.68	−0.38	0.08
Muscle Strength and Function												
MAC	−0.02	0.90	−0.34	0.09	0.26	0.09	−0.06	0.80	0.13	0.37	0.01	0.97
TSF	−0.02	0.89	−0.21	0.31	0.03	0.84	−0.13	0.57	0.03	0.84	0.42	0.04
CI	−0.07	0.68	0.16	0.46	0.07	0.67	−0.21	0.35	0.01	0.95	0.22	0.31
MAMA	−0.01	0.65	−0.34	0.10	0.30	0.04	−0.03	0.90	0.14	0.33	−0.15	0.48
AVI	−0.15	0.32	−0.01	0.97	0.21	0.19	−0.10	0.66	0.09	0.56	0.26	0.22
Handgrip Strength	0.02	0.87	0.01	0.96	0.13	0.37	0.01	0.97	−0.13	0.37	−0.06	0.78
TUG	−0.10	0.52	0.48	0.02	0.30	0.04	−0.17	0.44	−0.14	0.34	−0.11	0.60
Muscle Mass	−0.11	0.48	−0.08	0.69	0.23	0.12	−0.20	0.35	0.08	0.59	−0.34	0.09
R-leg-M	−0.13	0.39	−0.12	0.55	0.30	0.04	−0.22	0.31	0.12	0.42	−0.41	0.04
L-leg-M	−0.10	0.52	−0.14	0.50	0.11	0.46	0.09	0.67	0.13	0.37	0.10	0.64
R-arm-M	−0.05	0.75	−0.14	0.48	0.21	0.17	−0.29	0.16	0.06	0.66	−0.23	0.26
L-arm-M	−0.12	0.41	−0.14	0.51	0.22	0.14	−0.25	0.24	0.14	0.32	−0.20	0.32
Trunk	−0.09	0.53	0.05	0.80	0.23	0.12	−0.25	0.24	0.03	0.86	−0.35	0.08
Predictive equation (kg/m^2^)	−0.19	0.20	0.04	0.83	0.25	0.10	0.12	0.58	0.09	0.54	−0.01	0.98

Note: Data presented as coefficient (r); *p*-value (*p*) significant at <0.05. Non-Sarc, non-sarcopenia; BMI, body mass index; WHR, waist-hip ratio; Chol, cholesterol; MAC, mid-arm circumference; TSF, triceps skinfold thickness; CI, conicity index; MAMA, mid-arm muscle area; AVI, abdominal volume index; TUG, timed-up-and-go test; R-leg-M, right leg muscle; L-leg-M, left leg muscle; R-arm-M, right arm muscle; L-arm-M, left arm muscle.

**Table 6 diagnostics-12-02293-t006:** Logistic regression analysis using inflammatory marker cut-offs to determine predictive power in identifying those with sarcopenia.

Parameters	OR (95% CI)	*p*-Value
IL-6		
T1 (<8.5)	1.0	
T2 (8.51–10.42)	0.65 (0.19–2.21)	0.50
T3 (>10.42)	0.27 (0.06–1.18)	0.08
TNF-α		
T1 (<23.4)	1.0	
T2 (23.41–71.2)	3.56 (0.63–20.16)	0.15
T3 (>71.2)	5.85 (1.07–32.08)	0.04
CRP		
T1 (<1.6)	1.0	
T2 (1.61–4.86)	1.90 (0.53–6.76)	0.32
T3 (>4.86)	0.95 (0.24–3.81)	0.94

Note: Data presented an odd ratio (95 CI%). *p*-value significant at 0.05 and 0.01 level. IL-6, interleukin 6; TNF-α, tumor necrosis factor alpha; CRP, C-reactive protein; T, tertile.

## Data Availability

The study data are available from the corresponding author upon reasonable request.

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
