# Peer review of "Diagnostic Value of Inflammatory Markers in Elderly Arab Women with Sarcopenia"

_diagnostics, 2022, doi:10.3390/diagnostics12102293_

Round 1
Reviewer 1 Report
How was the two groups` size estimated? Was there an a priori sample size calculation? Why have the authors decided to have a control group (n=50) larger than the case group (n=26)?
In Table 3 and the paragraph above Table 3 - there are some abbreviations that are not explained. Please add legend. In Table 3, all data are presented as mean and SD, the footer note should be verified.
In Table 5 - in the footer there is an (partial) explanation for #, but # does not appear in table. Why there are missing correlation coefficient for the whole sample of participants?
What about the physical activity level of the participants?
The percentage of osteoporosis seems relatively low for the age of participants. How do authors explain? What about the current medication?
In my opinion, there should be more participants` data (medical, especially) presented that could have an impact on both muscle mass and inflammatory markers.
Author Response
Response to Reviewer 1
We thank the reviewers for their careful examination of the manuscript which substantially improved the scientific quality of our paper. Our point-by-point response is given below. Changes in the manuscript were tracked. Please note that the pages and line numbers mentioned in the reviewers’ comments refer to the original manuscript, whereas those in the authors’ reply refer to the revised manuscript.
- How was the two groups` size estimated? Was there an a priori sample size calculation? Why have the authors decided to have a control group (n=50) larger than the case group (n=26)?
Response: We thank the reviewer for this insightful comment and the following statement was added together with the data analysis subsection: Sample size was obtained from previous literature comparing inflammatory cytokines among Iranian adults with and without sarcopenia, noting increased CRP levels among cases with an effect size of 0.92. The calculation was done using G*Power (version 3.1) using α=0.05 and power 0.9, the estimated sample size was N=22 per group.
- In Table 3 and the paragraph above Table 3 - there are some abbreviations that are not explained. Please add a legend. In Table 3, all data are presented as mean and SD, and the footer note should be verified.
Response: We thank the reviewer for the scrutiny to detail. We spelled out all abbreviations and the legends were revised accordingly to clearly reflect how the data was presented.
- In Table 5 - in the footer there is a (partial) explanation for #, but # does not appear in the table. Why there are missing correlation coefficients for the whole sample of participants?
Response: We repeated the analysis in table 5 and found no significant associations using all participants. Hence, we presented only the associations according to sarcopenia status in the revised table. The footer was also modified accordingly.
- What about the physical activity level of the participants?
Response: We did not assess physical activity as we were mostly concerned with parameters needed to diagnose sarcopenia and its influence on inflammatory markers. We acknowledge that this information is useful since physical activity itself can influence inflammatory markers, but given the cross-sectional design, information on physical activity may provide more insights into perspective or interventional studies. We included this in the revised discussion, subsection on limitations.
- The percentage of osteoporosis seems relatively low for the age of participants. How do authors explain? What about the current medication?
Response: We totally agree with the reviewer with respect to the relatively low percentage of osteoporosis in the participants. Since the clinical information was provided by the elderly participants themselves and was not extracted from medical records, the accuracy of the medical history provided cannot be verified and is subject to recall bias. We have included this limitation in the revised discussion.
- In my opinion, there should be more participants` data (medical, especially) presented that could have an impact on both muscle mass and inflammatory markers.
Response: We agree with the reviewer that the clinical factors mentioned may influence muscle mass and inflammatory status of patients. However, given the small sample size, adjusting for these confounders will only increase the likelihood of type 2 errors. Nevertheless, we acknowledge the relevance of the comment and have accordingly included it in the revised discussion.

Reviewer 2 Report
Reviewer comments
|
Line |
Manuscript |
Comments |
|
Abstract |
the non-sarcopenia group (n=50) and the sarcopenia group (n=26) |
The authors should determine how the patients classified into 2 groups |
|
|
anthropometric measurements, skeletal..etc |
The authors should determine the tools used for assessment |
|
|
kin 6 (IL-6) was positively |
The authors did not refere to that at the methods |
|
|
inflammatory markers were significantly |
The conclusion should be written in the present tense |
|
Introduction |
14–33% (up to 68%) in institution |
Not scientific writing |
|
|
categorized as a geriatric syndrome |
Many sentences without refences |
|
|
Recent studies have shown that inflammatory cytokines could |
Recence studies ….and only one reference cited |
|
|
mass loss in postmenopausal |
English editing is mandatory in the whole manuscript |
|
Materials and methods |
This study design was a case-control study |
Please revise this point as the study is a cross sectional |
|
|
Subjects were divided into case and control groups |
Please determine how you classified your population |
|
|
Obstructive Pulm… |
Inappropriate use of capital and small letters |
|
|
Trained personnel took.. |
What is the type and level of training |
|
Results |
76 subjects, 50 women as a control group |
It is better to classify your groups as scropenia and non_sarcopenia as the term control gives an impression of totally healthy subjects |
|
|
Demographic Characteristics of Subjects According to Sarcopenia Status. |
Why you use capital letters without need |
|
|
Widowed divorced
|
Inappropriate use of capital and small letters |
|
|
0 (0.0)
|
0 |
|
Table 4 |
|
The correlation should include P value to be more informative and include sarcopenia group only which is the aim of the study |
|
Tables |
|
Any abbreviations in the tables should be defined at the foot notes |
|
Discussion |
this was in line with previous findings [15] |
Revise the English and grammar writing |
|
|
|
Strength and limiatations are not present |
|
|
|
The discussion writing is in need of improvement |
|
|
|
The conclusion is not consistent with the results |
|
|
|
The authors should determine why they included females only |
Author Response
Response to Reviewer 2
We thank the reviewers for their careful examination of the manuscript and appreciate the useful suggestions to improve the quality of our paper. Our point-by-point response is given below. Changes in the manuscript were tracked. Please note that the pages and line numbers mentioned in the reviewers’ comments refer to the original manuscript, whereas those in the authors’ reply refer to the revised manuscript.
Comments from the Editors and Reviewers:
- Abstract: the non-sarcopenia group (n=50) and the sarcopenia group (n=26). The authors should determine how the patients classified into 2 groups
Response: We thank the reviewer for raising this. The AWGS operational definition was used to stratify the participants. This is now mentioned in the revised abstract.
- Abstract: anthropometric measurements, skeletal..etc. The authors should determine the tools used for assessment.
Response: Tools used were now mentioned in the revised abstract.
- Abstract: kin 6 (IL-6) was positively. The authors did not refere to that at the methods.
Response: Details with regards to all inflammatory markers, including IL-6, are now mentioned in the revised methods.
- Abstract: inflammatory markers were significantly. The conclusion should be written in the present tense.
Response: It has been corrected.
- Introduction: 14–33% (up to 68%) in institution. Not scientific writing.
Response: It has been revised accordingly.
- Introduction: categorized as geriatric syndrome. Many sentences are without references.
Response: We thank the reviewer for taking note of this. Additional references were now inserted in the revised introduction and other parts of the manuscript, where needed.
- Introduction: Recent studies have shown that inflammatory cytokines could. Recent studies ….and only one reference cited.
Response: Additional references have been provided.
- Introduction: mass loss in postmenopausal. English editing is mandatory in the whole manuscript.
Response: This has been revised for English clarity.
- Materials and methods: This study design was a case-control study. Please revise this point as the study is cross-sectional.
Response: The design is indeed cross-sectional and is now mentioned in the revised methods.
- Materials and methods: Subjects were divided into case and control groups. Please determine how you classified your population
Response: The stratification of participants is now provided in detail in the revised study participants’ subsection.
- Materials and methods: Obstructive Pulm… Inappropriate use of capital and small letters.
Response: It has been corrected accordingly.
- Materials and methods: Trained personnel took... What is the type and level of training?
Response: This personnel was certified nurses working at the primary care centers where the study was conducted.
- Results: 76 subjects, 50 women as a control group. It is better to classify your groups as sarcopenia and non_sarcopenia as the term control gives an impression of totally healthy subjects.
Response: This point is well taken and as such, all groupings are now classified according to sarcopenia and non-sarcopenia groups.
- Results: Demographic Characteristics of Subjects According to Sarcopenia Status. Why you use capital letters without need.
Response: The use of capital letters in table titles are arbitrary and can be revised or left depending on the required journal formatting.
- Results: Widowed, divorced. Inappropriate use of capital and small letters.
Response: It has now been corrected.
- Results: 0 (0.0). 0
Response: It has been replaced with 0 only.
- Table 4: The correlation should include P value to be more informative and include sarcopenia group only which is the aim of the study.
Response: We believe the reviewer was referring to table 5 which showed the correlations. Indeed, upon the completion of missing associations the significant ones were observed only after stratification according to sarcopenia status. Hence, we removed the column for all participants and mentioned only the correlations according to groups. P-values were already indicated as footnotes where *denotes p<0.05 and **p<0.01. We believe this is better so the readers can immediately identify the flagged numbers as to which ones are significant. Including the exact p-values even for those that are not significant will make the table look busier than it already is.
- Tables: Any abbreviations in the tables should be defined at the foot notes.
Response: This has now been defined accordingly.
- Discussion: this was in line with previous findings [15]. Revise the English and grammar writing
Response: The discussion and the entire paper were revised accordingly for the right English usage.
- Discussion: Strength and limitations are not present.
Response: We thank the reviewer for this insightful comment. We have now included the strengths and limitations as a separate paragraph in the revised discussion.
- Discussion: The discussion writing is in need of improvement.
Response: The discussion was revised substantially.
- Discussion: The conclusion is not consistent with the results.
Response: The conclusion was revised accordingly to be compatible with the results presented.
- Discussion: The authors should determine why they included females only.
Response: This has been included in the revised discussion as part of the limitation section.

Round 2
Reviewer 1 Report
There are still some abbreviations not explained - like UGT.
English language should be checked and improved
Reviewer 2 Report
Accept in the present form